# A Model of Panic Buying and Workforce under COVID-19

**DOI:** 10.3390/ijerph192416891

**Published:** 2022-12-15

**Authors:** Guohua He, Zirun Hu

**Affiliations:** Economics and Management School, Wuhan University, Wuhan 430072, China

**Keywords:** COVID-19, panic buying, workforce, general equilibrium

## Abstract

Allowing there to be an undersupply of medical resources and infection amid the social workforce, this paper proposes a theory to show how panic buying is induced and how bad the workforce status could be. By developing a novel general equilibrium model, we find that for any retail price that is higher than the buyer’s reserve value, the buying competition will be induced and the medical resources supply will further be tightened. Moreover, if the transmission rate of COVID-19 surpasses the theoretical threshold that is proposed by this paper, the whole workforce in our simulated economy will inevitably be infected.

## 1. Introduction

Considerable efforts have been devoted to a better understanding of the economic cause of COVID-19 shock, mainly the ongoing researches over the trade-off between economic growth and mitigation of this pandemic, such as the works of Acemoglu et al. [1], Alvarez et al. [2], and other derivative studies. Acemoglu et al. [1] set the lockdown policy as an intermediary to connect the losses of lives and economic cost. Hence, an efficient frontier balancing both sides is presented by such methodology to conclude that the oldest group should be the priority of COVID policies. Be that as it may, few literatures have focused on the economic behaviors amid this pandemic, such as panic buying and violent turbulence of the labor force in the short term, and these missing pieces happen to be the important measures in response to the pre-crisis caution. Panic buying has become one of the severe behaviors under the zero-COVID policy; multiple coverage examples can be found at https://www.bbc.com/news/world-asia-china-60912846, accessed on 1 November 2022, https://edition.cnn.com/2022/03/28/china/shanghai-lockdown-china-covid-19-outbreak-intl-hnk/index.html, accessed on 1 November 2022, and https://edition.cnn.com/2022/06/10/china/china-covid-shanghai-mass-testing-intl-hnk/index.html, accessed on 1 November 2022.

A few studies closely related to our research have anatomized the determinants and influences of panic buying. Roberto et al. [3] discussed the flaws of the classical SIR model and pointed out a few limitations from both agent’s behavior and authorities’ policies. Chua et al. [4] proposed a specified health belief model to identify the determinants of panic buying, while Yuen et al. [5] based their study on the agent’s behavior and derived the psychological causes of panic buying. Engstrom et al. [6] observed the drastic panic buying behavior since the onset of COVID-19 by Australian consumption data; similarly, Yoshizaki et al. [7] surveyed Brazil’s situation and found that panic buying is increasing by income per capita. More theoretically, Mao et al. [8] developed an evolutionary game to discuss the choices between the public and government, and the results showed that the implementation of rumor-refutation strategy by government will significantly mitigate the intensity of panic buying. Chen et al. [9] also recognized that information amid the pandemic has a significant impact on an agent’s buying behavior, and the government may play an indispensable role to curb these behaviors. Moreover, Yuen et al. [10] developed a theoretical model explaining that non-coercive social influence, social norms, and observational learning directly influence one’s perception of scarcity, which could motivate panic buying. From the perspective of supply chain, Dulam et al. [11] and Herbon and Kogan [12] analyzed the possible impacts panic buying could bring to the supply chain. Some partial literatures include but are not limited to [13,14,15,16,17,18,19,20,21,22,23,24].

To explore these two pillar questions about our theme, we answered with a two-sector equilibrium model as the core and a full-fledged small SIR model as the framework. In the first part, we discuss the panic buying phenomenon by granting households and goods retailers, especially the mask retailers, the buying quotes and selling quotes, and then let the matching and bargaining mechanism mimic the market to settle the equilibrium. Without the loss of generality, a little rationing limit is introduced in order to echo the stylized facts. In the second part, the variation and properties of workforce dynamics are shown under the shroud of outbreak and infection, and conditions of a few extreme scenarios are provided for the sake of policy enlightenment. To our knowledge, this is the first paper that has such equations analyzed by a theoretical framework.

We underline that any retail price higher than the buyer’s reservation value would certainly arouse the buying competition on medical resources, resulting in intensifying market tightness. Moreover, given the high transmission ability of coronavirus, such as the Omicron variant, if the actual speed of infection surpasses the implied threshold in our model, the whole workforce in the simulated economy would eventually be infected.

There are also a few lines and boundaries about this paper. First, as most of the stylized facts show that people are commonly and intentionally hoarding resources under the negative shock wave of COVID-19, despite the supply side still running at a high capacity level, our research aims to explore the whole mechanism from the demand side instead of the supply side. Technically, the supply is assumed to be steady in the short term, while the variation of demand could be more volatile under the stimulus of buying competition. Second, the background of this paper is rooted in China and a few Asian countries, where authorities have implemented multiple regulations concerning the COVID-19 pandemic. Two of these rules are critically pivotal: one is wearing masks being mandatory in public, and another is that the masks cycling in the market should be factory manufactured and have to meet a high standard criterion; thus, any factors that violate such rules are ruled out in our study. Third, parts with complexity are excluded from our model in order to obtain analytical solutions for theoretical reasoning; hence, more details such as other DSGE sectors and computational heterogeneity can be added in the future.

## 2. Model Setups

In this section, we first lay down the basic roles of our two-sector equilibrium, which includes households and retailers, to analyze the arousal and the effect of panic buying; then, a full-fledged small SIR model is developed to characterize the variation of the actual workforce under the shock of COVID-19. To unambiguously describe the dissimilarities of agent’s decisions over different occasions, such as when medical resources are undersupplied or otherwise, the model starts with the outlines of households.

### 2.1. Households

The household sector is assumed to be heterogeneous over a continuum of measure such that $\int h^{i}di=1$, where *h* is the individual household and *i* denotes the *i*-th one. Take arbitrary *i* as an illustration; one needs to allocate their resources on consumption goods $C_{t}$ and surgical masks $M_{t}$, and also dedicate the labor supply $L_{t}$ if they are not infected (or detected) with COVID-19. The basic utility function of *i* takes the form of
$$\sum_{t=0}^{\infty }\beta^{t}u_{t}\left(C_{t},M_{t},L_{t}\right),$$
where $u_{t}$ is the household utility at time *t* and β<1 is the discount factor that transforms the future utilities into today; thus, a household aims to seek an optimal allocation path over ${\left\{C_{t},M_{t},L_{t}\right\}}_{t=0}^{\infty }$ to maximize its lifetime utility. Single period utility $u_{t}$ has a few properties such that
$$\begin{array}{cc} \partial u_{t}/\partial C_{t}>0, & \;\partial^{2}u_{t}/\partial C_{t}^{2}<0;\\ \partial u_{t}/\partial M_{t}>0, & \;\partial^{2}u_{t}/\partial M_{t}^{2}<0;\\ \partial u_{t}/\partial L_{t}<0, & \;\partial^{2}u_{t}/\partial L_{t}^{2}<0, \end{array}$$
these properties also suggest that *i* is risk aversion. We assume that the above-specified utility function is identical across the whole household sector.

To be precise, if *i* is neither infected nor diagnosed at time *t*, they shall earn the income from both labor supply at competitive wage $W_{t}$ and the payback interest $R_{t}$ of bond $B_{t-1}$. With all the income *i* has, they need to choose the quantities of non-durable goods $C_{t}$ for consumption, saving $B_{t}$ for the next period, and surgical masks $M_{t}$ to keep away from COVID-19 infections given the maximization of the lifetime utility. In order to deliberately characterize the stylized facts during the onset of this pandemic, a few assumptions are in made.

**Assumption** **1.**
*Without the loss in generality, assume that*
*(i)* 
*Resources for producing masks are relatively steady;*
*(ii)* 
*Surgical mask depreciates at the rate of δ;*
*(iii)* 
*Only*

$1-\rho_{t}$

*fraction of mask demand (*

$M_{t+1}^{*}$

*) in each household is satisfied;*
*(iv)* 
*Mask restocking is only for balancing the mask consumption.*



Assumption 1–(i) says that the surgical mask $M_{t}$ is durable but depreciating at rate δ in each period. If $M_{t}$ are the masks *i* obtained at the beginning of time *t*, then the depreciation $\delta M_{t}$ equals to the masks *i* used in period *t*, so that $\left(1-\delta \right)M_{t}$ quantifies the size of masks remaining unused from *t* to t+1. The way that households use masks across periods resembles peeling an onion. For instance, given a certain number of masks $M_{t}$ at time *t*, the depreciation rate is δ for the masks used at time *t*; so, the remaining number at t+1 should have $\delta \left(1-\delta \right)M_{t}$ to be used and ${\left(1-\delta \right)}^{2}M_{t}$ stays unused. Forward iterate this for *n* periods, the used and unused masks are given by $\left(1-{\left(1-\delta \right)}^{n}\right)M_{t}$ and ${\left(1-\delta \right)}^{n}M_{t}$; so, if *n* is sufficiently big, then all the masks given from time t are consumed. To fully cover the “mask shortage” in reality, the total output of masks in mainland China of 2019 is 5 billion according to China’s industrial statistics, where the surgical masks comprise 54% of this number, which is 2.7 billion. However, regarding the mask demand, assuming one man consumes one mask for a day from China’s secondary industry, transportation industry, as well as medics, the total number of masks used was 238 million per day in 2019 according to *The Fourth National Economic Census* and *China Health Statistics Yearbook 2019*, implying that the surgical mask stock only maintains for 11 days (27/2.38≈11) if the workforce from preceding industries decides to return to work during the epidemic, which most of them did from 20 January. Meanwhile, more than half of the mask factories were in a state of stagnation due to the Chinese new year. Assumption 1–(ii) states that only relatively low fractions of masks can be purchased by *i* from the market in each period, suggesting that masks could be undersupplied and the stabilizations are highly dependent on the market statutes and related policies. Take Shenzhen as an example, roughly 10 million netizens take a lottery draw every two days for 20 thousand surgical masks freely supplied by the government, while the surgical masks have been sold out in most of the pharmacies, and even in the city with multiple local surgical mask factories. To survive the pandemic as best as possible, *i* manages to restock a certain number of masks that is in line with their economic decisions in each period; this is the intuition behind Assumption 1–(iii). Mathematically, the transition law of mask without shortage can be expressed as
$$M_{t+1}=\left(1-\delta \right)M_{t}+O_{t}$$
where $O_{t}$ denotes the restock at time *t* according to Assumption 1–(i,iii). Since we acknowledge the possibility of mask undersupply, a wedge is introduced to decay the mask accumulation such that
(1)$$M_{t+1}=\left(1-\delta \right)M_{t}+O_{t}-\Delta_{t},$$
where $O_{t}-\Delta_{t}$ is the “mask shortage gap”, which helps pin down the exact number of masks *i* should restock at time *t*. According to Assumption 1-(ii), *i* can only restock $1-\rho_{t}$ fraction of masks at most—that is, $\Delta_{t}\leq \rho_{t}\left[\left(1-\delta \right)M_{t}+O_{t}\right]$ with $\rho_{t}<1$; hence, (Equation 1) is further transformed into an inequality constraint as follows:(2)$$M_{t+1}\geq \left(1-\rho_{t}\right)\left[\left(1-\delta \right)M_{t}+O_{t}\right].$$
Note that constraint (Equation 2) binds whenever masks are insufficient to *i*, and $\Delta_{t}=0$ when the mask shortage situation disappeared. Moreover, with the help of Equations (Equation 1) and (Equation 2), we can develop the budget constraint for *i* such that
(3)$$C_{t}+B_{t+1}+O_{t}-p_{t}\Delta_{t}=W_{t}L_{t}+R_{t}B_{t},$$
where $O_{t}-p_{t}\Delta_{t}$ is deemed as the authentic mask shortage gap, which stays as a real variable in (Equation 3), as we will verify that $p_{t}>1$ when mask supply is relatively low and $p_{t}=1$ otherwise. To gain more insight, first rewrite (Equation 2) into
$$O_{t}=\frac{M_{t+1}}{1-\rho_{t}}-\left(1-\delta \right)M_{t}$$
and then plug it into (Equation 3) with the condition of $\Delta_{t}=\rho_{t}\left[\left(1-\delta \right)M_{t}+O_{t}\right]$; hence, we obtain
(4)$$C_{t}+B_{t+1}+\frac{1-\rho_{t}p_{t}}{1-\rho_{t}}M_{t+1}=W_{t}L_{t}+R_{t}B_{t}+\left(1-\delta \right)M_{t}.$$

As can be concluded by (Equation 4), term $\left(1-\rho_{t}p_{t}\right)/\left(1-\rho_{t}\right)$ represents the willingness of *i* to buy a mask: the expense for purchasing a mask is lower than the value of the mask itself if $\left(1-\rho_{t}p_{t}\right)/\left(1-\rho_{t}\right)<1$, indicating that it stimulates buying of the mask.

If *i* is infected and also diagnosed, however, they will receive medical treatment at a hospital, and it temporarily blocks their ability to work. Moreover, we also assume that *i* will not need a mask during their time in hospital. Under such a case, the utility function of *i* is given by
$$\sum_{t=0}^{\infty }\beta^{t}u_{t}\left(C\right),$$
and the associated budget constraint is
$$C_{t}+B_{t+1}=R_{t}B_{t-1}.$$

### 2.2. Retailers

There is a continuum of firms of measure one in the model economy; each retailer possesses the Cobb–Douglas technology that selects labor as the sole ingredient for production, i.e.,
(5)$$Y_{t}=A_{t}L_{t}^{\alpha },$$
where we drop the individual subscripts *i* as retailers and outputs are identical. The output is divided into partition $\left\{C_{t},O_{t}-p_{t}\Delta_{t}\right\}$ in each period with $C_{t}\cap \left(O_{t}-p_{t}\Delta_{t}\right)=⌀$ and
(6)$$Y_{t}=C_{t}+\int O_{t}-p_{t}\Delta_{t}di,$$
that is, consumption goods and masks are simultaneously produced by retailers. Note that we are more leaning to let the competitiveness settle the result of Equation (Equation 6) rather than specify the proportion for consumption goods $C_{t}$ or aggregate mask $\int O_{t}-p_{t}\Delta_{t}di$ in the output. The only production cost retailers are faced with is the wage paid to the labor supply from households; hence, the optimal condition is given by
(7)$$W_{t}=\alpha A_{t}\frac{Y_{t}}{L_{t}}.$$

Moreover, in order to focus on the following matching frictions between households and retailers, we assume $A_{t}\equiv 1$.

### 2.3. Matching and Bargaining

Matching and bargaining between households and mask retailers as the key factor of our model is detailed in this subsection. There are two reasons for us to build up this mechanism: first, as the assumption stated, only a fraction of mask demand in each household is met; second, matching and bargaining offer a practical and tractable way to endogenize the mask shortage situation, and further helps us analytically explore how shortage $\rho_{t}$ interacts with infected household.

To underline the heterogeneity in households, we assume that every household may reserve a different valuation on masks such that $u_{M}\in \left[\underline{u_{M}},\bar{u_{M}}\right]$ with distribution G(u) for $\bar{u_{M}}>\underline{u_{M}}$, where we choose the partial derivative of utility *u* to mask *M* as the valuation criterion and $u_{M}>0$ given any level of *M* due to utility function *u* satisfying Inada conditions. Moreover, given the marginal cost of providing a mask from firms at time *t*, $mc_{t}$, we let $\bar{u_{M,t}}>mc_{t}\geq \underline{u_{M,t}}$ for ruling out the case of market freeze. Likewise, we further define $\mu_{t}$ as the reservation value such that households are willing to buy masks if and only if $u_{M,t}\geq \mu_{t}$. Above the reservation value, a buying price offer is derived from a household, $p_{t}=p_{t}\left(u\right)$; the specific form of this buying price offer will be discussed below.

In line with the buying fractions $\rho_{t}$ in (Equation 1), the arrival rates for households (buyers) meeting retailers (sellers) are equivalent to the fractions that meet their restocking plan on average—that is,
$$\kappa_{t}=\frac{O_{t}-p_{t}\Delta_{t}}{O_{t}}=\left(1-\rho_{t}\right)-p_{t}\cdot \frac{\left(1-\delta \right)\rho_{t}\left(1-\rho_{t}\right)M_{t}}{M_{t+1}-\left(1-\delta \right)\left(1-\rho_{t}\right)M_{t}}<1,$$
where we used the expressions of $O_{t}$ and $\Delta_{t}$. Denote $V_{H}$ as the value function for the household who preserves valuation $u_{M,t}$; hence, the value this kind of household places on buying masks is given by
(8)$$V_{H,t}=\kappa_{t}e^{-\kappa_{t}}\left[u_{M,t}-p_{t}\left(u\right)\right]+e^{-\kappa_{t}}\beta E_{t}V_{H,t+1},$$
where β is the discount factor. The first term on the RHS of Equation (Equation 8) is the probability of a household obtaining a mask supply $P\left\{X=1\right\}=\kappa_{t}e^{-\kappa_{t}}$ over Poisson distribution, while the second term says, otherwise, it is at probability $P\left\{X=0\right\}=e^{-\kappa_{t}}$, where $V_{H,t+1}$ stands for the value function of the next period.

Unlike the households, retailers face numerous buying quote ubiquitously due to mask shortages; hence, the arrival rates for retailers (sellers) to meet households (buyers) are constantly number 1. Moreover, it is useful to consider the whole retailer sector as a representative agent due to lacking of dissimilarities across retailers; then, the associated Bellman equation is given by
(9)$$\begin{array}{cc} V_{F,t} & =E_{t}\left[\max e^{-1}\cdot \left\{p_{t}\left(u\right)-mc_{t},V_{F,t}\right\}+e^{-1}\beta V_{F,t+1}\right]\\ \; & =e^{-1}\left[\int_{0}^{\infty }\max\left\{p_{t}\left(u\right)-mc_{t}-V_{F,t},0\right\}dG\left(u\right)+V_{F,t}+\beta E_{t}V_{F,t+1}\right]\\ \; & =e^{-1}\left[\int_{\mu_{t}}^{\infty }p_{t}\left(u\right)-mc_{t}-V_{F,t}dG\left(u\right)+V_{F,t}+\beta E_{t}V_{F,t+1}\right]. \end{array}$$
These two Bellman equations can also be interpreted into continuous time form. Suppose the interval of time is sufficient small; then, Equation (Equation 8) becomes
$$\begin{array}{cc} V_{H}=\kappa e^{-\kappa t}\left[u_{M}-p\left(u\right)\right]t+e^{-\kappa t}e^{-rt}V_{H}\; & \;\Rightarrow \;\lim_{t\to 0}\frac{1-e^{-(r+\kappa )t}}{t}V_{H}=\lim_{t\to 0}\kappa e^{-\kappa t}\left[u_{M}-p\left(u\right)\right]\\ \; & \;\Rightarrow \;rV_{H}=\kappa \left[u_{M}-p\left(u\right)-V_{H}\right], \end{array}$$
where *r* is net interest. Likewise, Equation (Equation 9) gives us
$$\begin{array}{cc} V_{F}=E\left[\max e^{-t}\cdot \left\{p\left(u\right)-mc,V_{F}\right\}t+e^{-t}e^{-rt}V_{F}\right]\; & \;\Rightarrow \;\lim_{t\to 0}\frac{1-e^{-(r+1)t}}{t}V_{F}=\lim_{t\to 0}e^{-t}\left[\max\left\{p\left(u\right)-mc-V_{F},0\right\}+V_{F}\right]\\ \; & \;\Rightarrow \;rV_{F}=\int_{0}^{\infty }\max\left\{p\left(u\right)-mc-V_{F},0\right\}dG\left(u\right). \end{array}$$

Nash bargaining is employed to assume the endogenous retail price. Obviously, the surplus of type $u_{M,t}$ households is $u_{M,t}-p_{t}\left(u\right)-V_{F,t}$, while the surplus of retailers is $p_{t}\left(u\right)-mc_{t}-V_{H,t}$; then, the bargaining problem is
$$p_{t}\left(u\right)=\mathrm{argmax}{\left[u_{M,t}-p_{t}\left(u\right)-V_{F,t}\right]}^{\theta }{\left[p_{t}\left(u\right)-mc_{t}-V_{H,t}\right]}^{\left(1-\theta \right)},$$
where the bargaining power of households is given by θ, and the solution to this bargaining problem implies that
(10)$$p_{t}\left(u\right)=\theta \left(mc_{t}+V_{F,t}\right)+\left(1-\theta \right)\left(u_{M,t}-V_{H,t}\right).$$

The following lemma helps summarize all the related properties for the sake of our discussion.

**Lemma 1.** *In the neighbourhood of steady state, the following conditions hold if and only if *$u_{M,t}=\mu_{t}$:
*(i)* *The retail price is equal to the reservation value,*$p_{t}\left(\mu \right)=\mu_{t}$;*(ii)* *The value function of households and retailers are *$V_{H,t}=0,V_{F,t}=\mu_{t}-mc_{t}$, respectively.

**Proof.** Given the fact that $u_{M}=\mu$, the integral in Equation (Equation 9) implies $p\left(\mu \right)-mc-V_{F}=0$; hence, we have $p\left(\mu \right)=mc+V_{F}$. Substituting this condition into Equation (Equation 10) gives $p\left(\mu \right)=\mu -V_{H}$, and the steady state of Equation (Equation 8) is

$$\left[1-\left(\beta +\kappa \right)e^{-\kappa }\right]V_{H}=\kappa_{t}e^{-\kappa_{t}}\left[\mu -p\left(\mu \right)-V_{H}\right].$$
The preceding equation implies that $V_{H}\left(\mu \right)=0$ and p(μ)=μ. Moreover, by condition of $p\left(\mu \right)=mc+V_{F}$, we therefore have $V_{F}\left(\mu \right)=\mu -mc$. □

One thing that should be noted about Lemma 1 is that all the surplus in the trade goes to retailers when the equilibrium price is $\mu_{t}$. Consider two cases for the understanding of this result. First, assume that $u_{M,t}=\mu_{t}<p_{t}\left(\mu \right)$. The steady state implies the value of type μ is negative, i.e., $V_{H}=\mu -p<0$; then, households would utilize their bargaining capacity to prevent a negative value from occurring, this effort is denoted μ≥p. Second, assume $u_{M,t}=\mu_{t}>p_{t}\left(\mu \right)$, a type μ household would prefer trading with retailers since their surplus is increasing in mask purchasing, i.e., $V_{H}=\mu -p>0$. This circumstance gives rise to buying competition and eventually increases the bargaining capacity of firms, resulting in the soaring of retail price until $\mu =p,V_{H}=0$. These two cases have described the mechanism behind Lemma 1, despite this representing the worst-case scenario for the households.

To obtain more insight, we next compute the steady states of two value functions alongside retail price. In doing so, first, inserting (Equation 10) into (Equation 8) yields the value of households for all $u_{M}\geq \mu$:(11)$$V_{H}=\frac{\theta \kappa e^{-\kappa }}{1-\left(\beta +\kappa \right)e^{-\kappa }+\theta \kappa e^{-\kappa }}\left(u_{M}-mc-V_{F}\right).$$
Likewise, substituting (Equation 10) into (Equation 9) yields
(12)$$\mu -mc=\frac{1-\left(\beta +\kappa \right)e^{-\kappa }}{1-\left(\beta +\kappa \right)e^{-\kappa }+\theta \kappa e^{-\kappa }}\cdot \frac{\left(1-\theta \right)e^{-1}}{1-\left(\beta +1\right)e^{-1}}\int_{\mu }^{\infty }u_{M}-\mu dG\left(u\right),$$
where we use $V_{F}=\mu -mc$. Moreover, combine (Equation 8) and (Equation 10) to obtain
(13)$$p=\frac{\theta \left(1-\beta e^{-\kappa }\right)\mu +\left(1-\theta \right)\left[1-\left(\beta +\kappa \right)e^{-\kappa }\right]u_{M}}{1-\left(\beta +\kappa \right)e^{-\kappa }+\theta \kappa e^{-\kappa }}.$$

There are two aspects we would like to underline.

**Remark 1.** *At steady state or in the neighborhood of steady state, a household’s value $V_{H}$ is increasing in bargaining power θ and the marginal utility on mask, while it is decreasing in firm value $V_{F}$ and marginal cost mc. In comparison, firm value $V_{F}$ is increasing in its bargaining power 1−θ and marginal utility u but decreasing in reservation value μ*.

**Remark 2.** *Given the reservation value μ, retail price p is increasing in a household’s marginal utility u; hence, p(u)>p(μ)=μ holds for every u>μ*.

Remark 2 is crucial to the mask shortage characterization in our model. Assume the reservation value is equal to 1, μ=1; then, every valuation above the reservation should be p(u)>p(1)=1, and the authentic price of mask for non-reservation household is lower than 1 due to $\left[1-\rho_{t}p\left(u\right)\right]/\left(1-\rho_{t}\right)<1$. Moreover, according to Inada conditions, the marginal utility should be much bigger than 1 when masks are undersupplied, implying p(u)≫1 and $\left[1-\rho_{t}p\left(u\right)\right]/\left(1-\rho_{t}\right)\ll 1$, which would further trigger “panic buying” and intensify market tightness. Given this logic, we obtain Proposition 1.

**Proposition 1.** *Given the critical value p(μ)=μ=1, any valuation above the reservation echoing the authentic price of masks of less than 1, i.e., $\left[1-\rho_{t}p\left(u\right)\right]/\left(1-\rho_{t}\right)<1$, will further escalate the mask shortage circumstance*.

**Proof.** Omitted. □

A few things about Proposition 1 are worth commenting on; however, we first need to discuss the mask shortage equilibrium. Figure 1 displays the basic mask shortage equilibrium trends along with the changes of depreciation rate, where we set parameter δ from 0.54 to 0.6. As can be seen, the demand of masks rises when the pandemic arouses: for one thing, the restocking for an arbitrary household in each period, denoted by O−pΔ, increases from 0.54 to 0.57; for another, the marginal utility roughly rises from 1 to 1.1 due to the mask stock *M* running low, resulting in the pushing up of retail price p(u) and the downfall of authentic price $p^{*}$ according to Equation (Equation 13). Moreover, the fraction of masks that households could not restock, ρ, increases from 0 to 0.3961, where the shortage gap is widened by the excessive demand. One remark is on the order. Apparently, the fraction of masks that households could not restock slides toward 0 as the retail price and authentic price approach 1, where the mask market is Walrasian, and the arrival rates for households to meet retailers become 1 as ${\kappa |}_{\rho =0,p=1}=1$. Alternatively speaking, along with the authentic price $p^{*}$ dropping down from 1 household realizing that the mask would be cheaper than its fixed value 1 when all other agents expect the market to become tightened based the soaring demand of masks, then households are triggered to a buying competition, which as the original cause of the mask shortage, i.e., an increase in ρ also increases the mask restocking quantities, O−pΔ. However, the illustration is not straightforward since the panic buying effect under this scenario is always mixed with the effect of depreciation rate, indicating that more details should be characterized to explain Proposition 1.

Figure 2 is supplementary to Figure 1. The left panel anatomizes Proposition 1 in concreteness, where the blue line is the function of $u_{M}$ and the green line is p(u), and the equilibrium for pair $\left(u_{M},p\right)$ is in sequence from nodes $1^{\prime }$ to $4^{\prime }$. Set an initial state on the green line, which gives $\left(u_{M,1^{\prime }},p_{1^{\prime }}^{*}\right)$ from the retailers’ supply, where $p^{*}$ is defined as (1−ρp)/(1−ρ). However, since the price level $p_{1^{\prime }}^{*}$ only represents the fraction of mask supply that can be met by the demand on the blue line, i.e., state $\left(u_{M,2^{\prime }},p_{2^{\prime }}^{*}\right)$ from households, then retailers would re-optimize their authentic prices according to the demand $u_{M,2^{\prime }}$, which drags the supply state from node $1^{\prime }$ to node $3^{\prime }$. At the time that retailers reset their price $p^{*}$ in node $3^{\prime }$, the demand state from households deviates from node $2^{\prime }$ to node $4^{\prime }$, and the increment of masks $\Delta M=M\left(u_{M,4^{\prime }}\right)-M\left(u_{M,2^{\prime }}\right)$ is equal to the quantities of panic buying. Moreover, from Equation (Equation 9), we know that retailers are more willing to sell only when *p* is greater than the marginal cost mc, and the marginal cost in our model is constantly 1, as no nominal rigidities are embedded in the retailer sector; hence, the case of $p^{*}>1$ is ruled out. Meanwhile, we also omit the case of $p^{*}=1$, where the retail price is totally fixed and marginal utility stays in constant at equilibrium.

The right panel of Figure 2 gives an illustration of retail price shifting. To be more specific, we shift the retail price curve p(u) downward to $p{\left(u\right)}^{\prime }$ by lifting the reservation value μ from 1 to 1.01, echoing the case that households suddenly consider masks to be more valuable than ever. However, at the same time, we leave the utility function curve $u_{M}$ unchanged. As the rising of retail price *p*, i.e., the decreasing of authentic price $p^{*}$, the utility function $u_{M}$ is intersected at nodes $M_{1}$ and $M_{2}$, where $M_{1}$ and $M_{2}$ denote the exact quantities of the masks households possessed under two different retail prices, and $M_{2}>M_{1}$ holds as $u_{M}$ is decreasing in *M*. Obviously, $M_{2}-M_{1}$ as an embodiment of panic buying from mask price rising.

### 2.4. Hospital and Quarantine

Figure 3 gives a brief road map of the COVID-19 transmission. At the beginning of each period, buyers leave home to participate in manufacturing at retailers; meanwhile, a necessary test for infection is held before buyers start their work. If a buyer is negative to infection, they may continue to work but with a certain probability of contracting the virus and then becoming infected; after that, they return from work and back to home at the end of the period. If a buyer’s test result is positive, they will be considered as a diagnosed case and sent to hospital (or quarantine zone) for medical treatment, where the patient has a certain probability to recover and be free to leave. One assumption that should be underlined is that the masks are not necessary as long as the buyer stays in hospital.

Next, we transform the preceding transmission into mathematical details. Assume the total working force in time *t* is $N_{t}+E_{t-1}$, where $N_{t}$ denotes the newly emerging workforce, including the ones who have recovered from medical treatment at t−1, not contacted or been infected at t−1, and those of the debuted labor force in the current period, while $E_{t-1}$ are the negative cases to test from t−1 to *t*, who might have been infected during work. As stated, $E_{t-1}$ would be diagnosed as infected with the probability of σ at the beginning of time *t*. The rest of those are entering daily work, and the probability of the debuted labor force be diagnosed is 0. Within the work part, the workforce may directly expose themselves to the incubated with probability of *c*, and become infected with the probability of η. Therefore, the transition law of $E_{t}$ is as follows:(14)$$E_{t}=c\eta N_{t}+\left(1-\sigma \right)E_{t-1},$$
where the first term on the RHS denotes the labor workers who make contact and eventually become infected, and the second term is the existing group that does not get diagnosed at time *t*.

For these who are diagnosed, $\sigma E_{t-1}$, the associated medical treatments are offered by the hospital, and the probability of restoring a patient to health is γ. Hence, the dynamic equation for the diagnosed patient $Q_{t}$ is
(15)$$Q_{t}=\sigma E_{t-1}+\left(1-\gamma \right)Q_{t-1},$$
where the first term on the RHS is the new cases, and the second is the cases that remain uncured.

Last but not the least, the change of the size of recovered patient $J_{t}$ is equal to the number of the diagnosed cases that get cured, i.e.,
(16)$$J_{t}=\gamma Q_{t-1}.$$

Moreover, the summation of $\left\{N_{t},E_{t},Q_{t},J_{t}\right\}$ should equal the size of households—that is,
(17)$$N_{t}+E_{t}+Q_{t}+J_{t}=H_{t},$$
and $H_{t}$ is normalized to 1 for simplicity. Given equations (Equation 14)–(Equation 17), we obtain the following Lemma:

**Lemma 2.** $E_{t}$ is
*(i)* *Taking the whole labor force, expect the ones who are diagnosed only when*cη=(1−σ)/σ and σ>1/2;*(ii)* *Increasing in contact rate c, while decreasing in uncured rate *1−γ.


**Proof.** Substituting Equations (Equation 15)–(Equation 17) into (Equation 14) and by forward iteration yields

(18)$$E_{t}=\psi^{t-1}E_{0}+\left(1-\frac{1}{1+c\eta }\right)\sum_{i=0}^{t-1}\psi^{i}\left(1-Q_{t-1-i}\right)=\left(1-\frac{1}{1+c\eta }\right)\sum_{i=0}^{t-1}\psi^{t-1-i}\left(1-Q_{i}\right),$$
where the parameter ψ is the short symbol for 1/(1+cη)−σ. The first term on the RHS is negligible given $E_{0}\in \left[0,1\right)$ and when *t* is sufficient big. Now, let ψ=0—that is,
$$c\eta =\frac{1-\sigma }{\sigma },$$
the probability for a household eventually getting infected is equal to the odds in favor of not getting diagnosed. Therefore, the equation of $E_{t}$ turns into
$$E_{t}=\left(1-\sigma \right)\left(1-Q_{t-1}\right),$$
indicating that all households are negative to test expect for the newly diagnosed and quarantined ones.

Moreover, given the specification of (Equation 15), forward iterating gives
(19)$$Q_{t}=\sigma \sum_{j=0}^{t-1}{\left(1-\gamma \right)}^{i}E_{t-1-j}=\sigma \sum_{j=0}^{t-1}{\left(1-\gamma \right)}^{t-1-j}E_{j}.$$

Combining (Equation 18) and (Equation 19) yields
$$\begin{array}{cc} E_{t} & =\left(1-\frac{1}{1+c\eta }\right)\sum_{i=0}^{t-1}\psi^{t-1-i}\left[1-\sigma \sum_{j=0}^{i-1}{\left(1-\gamma \right)}^{i-1-j}E_{j}\right]\\ \; & =1-\frac{\sigma }{1-\frac{1}{1+c\eta }+\sigma }-\sigma \sum_{i=0}^{t-1}\psi^{t-1-i}\sum_{j=0}^{i-1}{\left(1-\gamma \right)}^{i-1-j}E_{j}\\ \; & =1-\frac{\sigma }{1-\frac{1}{1+c\eta }+\sigma }-\sigma \sum_{i=0}^{t-1}\psi^{t-1-i}{\left(1-\gamma \right)}^{i-1}\sum_{j=0}^{i-1}{\left(1-\gamma \right)}^{-j}E_{j}\\ \; & =1-\frac{\sigma }{1-\frac{1}{1+c\eta }+\sigma }-\sigma \sum_{i=0}^{t-2}{\left(1-\gamma \right)}^{-i}E_{i}\sum_{j=0}^{t-1}\psi^{t-1-(i+j)}{\left(1-\gamma \right)}^{(i+j)-1}, \end{array}$$
where in the last step we used Abel’s summation as well as the conditions for i,j such that t−1>i+j>1 and j≥2 when i=0. Define coefficient term $\phi \left(i\right)=\sum_{j=0}^{t-1}\psi^{t-1-(i+j)}{\left(1-\gamma \right)}^{j-1}$; then, the last dynamic summation of $E_{t}$ is simplified into
$$E_{t}=1-\frac{\sigma }{1-\frac{1}{1+c\eta }+\sigma }-\sigma \sum_{i=0}^{t-2}\phi \left(i\right)E_{i}.$$

Obviously,
$$\frac{\partial E_{t}}{\partial c}=\frac{\frac{\sigma }{{\left(1+c\eta \right)}^{2}}}{{\left[1-\frac{1}{\left(1+c\eta \right)}+\sigma \right]}^{2}}-\sigma \sum_{i=0}^{t-2}\frac{\partial \phi (i)}{\partial c}E_{i}>0$$
and
$$\frac{\partial E_{t}}{\partial (1-\gamma )}=-\sigma \sum_{i=0}^{t-2}\frac{\partial \phi (i)}{\partial (1-\gamma )}E_{i}\leq 0$$
hold due to ∂ϕ(i)/∂c<0 and ∂ϕ(i)/∂(1−γ)>0. □

Lemma 2 presents the basic properties that the COVID-19 transmission embodies. We next conclude two remarks for the intuitive interpretation of this Lemma.

**Remark 3.** 
*Lemma 2–(i) describes the worst-case scenario of virus spread; the total number of labor forces, expect for the diagnosed and quarantined ones, are potentially infected as cη=(1−σ)/σ. Normally, the current negative cases $E_{t}$ will fit the parameter condition of cη<(1−σ)/σ, i.e., the probability of labor force becoming infected is lower than the odds in favor of not getting diagnosed. From Lemma 2–(ii), we also know that $E_{t}$ is increasing in contact rate c; hence, the parameter inequality morphs into equality alongside the uprising of contact rate. Intuitively, we can simply rewrite the parameter condition into*

cη>cη·σ=1−σ,

*which implies that the speed of infection, as measured by cη, is faster than the varying of existing negative cases, 1−σ, as the contact rate increases, so that the truly uninfected ones are sliding into negative cases and being fully crowded out.*


**Remark 4.** 
*Lemma 2–(ii) has exposited how the parameters of contact rate and uncured rate influence the quantities of negative cases. As a consequence, negative cases are increasing in contact rate c, while decreasing in uncured rate 1−γ. The message of the former one is not hard to comprehend: the more people exposed to the virus, the less uninfected cases there will be. As a matter of fact, Wuhan city has paid the price for indulging its citizens in holding gatherings, such as traditional large banquets and annual meetings including those of enterprises and local government, until the city was locked down on 23 January. Meanwhile, the disclosure of the number of infected was “technically censored” by an unauthorized medical emergency guide from Wuhan municipal health commission during 10–15 February. At the end of 15 March, the official number of diagnosed was roughly 50 thousand. However, the intuition behind the latter fact was slightly implicit. The uncured rate defined in our model is the chance that the diagnosed patients are not fully recovered from the virus even after the medical treatment, which also says that the uncured ones have to be kept in quarantine and away from the exposure of the virus again. Therefore, the larger the uncured rate 1−γ, the more people will become quarantined, and the less people will be exposed to the virus and infected in the future. This theoretical claim is in line with the following facts: (1) As the epicenter, Wuhan City and Hubei Province declared national emergencies as well as board lockdowns on 23 January at 10 a.m. to contain the spread of COVID-19; meanwhile, a nationwide self-quarantine was asked of the citizens who were in the other 31 provinces and autonomous regions. (2) All the schools including universities in mainland postponed their spring semester regarding this epidemic, and most of the employees from different industries were not allowed to return to work until early March. (3) Closed-off management has been implemented in the residential communities of first-tier and second-tier cities.*


## 3. Conclusions

In our analysis, pandemic shock is shown to affect an agent’s trade behavior from both the demand side and the supply side, and a more detailed mechanism of workforce variation amid this pandemic has developed according to the specific small SIR framework in our model. We found a few broad conclusions that explain the stylized facts to some extent:

First, by setting the trade between households and retailers through matching and bargaining, one can always find that a buyer’s surplus is increasing in their bargaining power and marginal utility on masks, while it is decreasing in seller’s surplus and the marginal cost of producing the masks. In contrast, the seller’s surplus is increasing in their bargaining power and buyer’s marginal utility but decreasing in buyer’s reservation value over masks (buyer’s bottom line for this trade). Thus, the statutes of a buyer’s marginal utility and reservation value become the heart of equilibrium.

Second, given the situation that only a fraction of masks ($\rho_{t}$) can be restocked in each period for every household, any retail price higher than the reservation value would arouse the buying competition on masks, resulting in escalation of the mask shortage circumstance.

Third, if the speed of infection (cη in our model) is faster than the variation of existing negative COVID test cases (1−σ), then the truly uninfected ones could be fully crowded out and the whole workforce would be potentially infected. However, this trend could be brought to a halt if the size of the quarantined group is sufficiently large despite it certainly reducing the productivity.

Policy recommendations include two main aspects. First, according to our theory (Proposition 1), authorities could take two effective strategies to fight this panic buying behavior amid the COVID-19 pandemic. One is putting a cap on the retail price of masks. The main reason why this would work effectively is because panic buying only happens when the mask price exceeds the buyer’s reserve value, and the high mask price could also accelerate the buying competition. Thus, any price cap less than the buyer’s reserve value could eradicate this vicious circle from the beginning. Another strategy is mask rationing, which could achieve the same result as the former one but is relatively hard to implement. Second, although the paper pointed out that the whole workforce infection could be brought to a halt if the quarantine size is sufficiently big, the trade-off here is the downfall of productivity. A feasible solution is to separate the whole workforce by age and job productivity, then assign most of the elderly and low-productivity workers who are diagnosed or potentially infected into quarantine and keep the remaining as usual as possible. This policy recommendation is also similar to the point in Acemoglu et al. [1].

## Figures and Tables

**Figure 1 ijerph-19-16891-f001:**
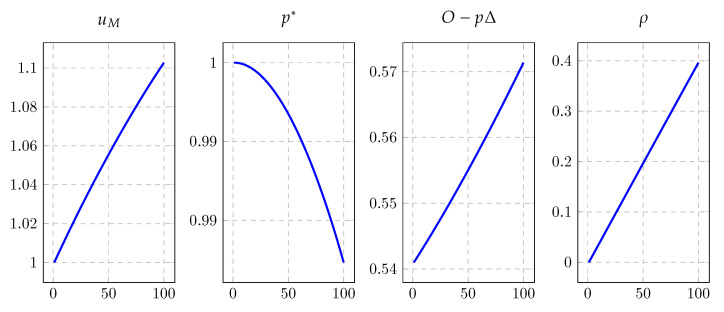
Equilibrium of mask shortage.

**Figure 2 ijerph-19-16891-f002:**
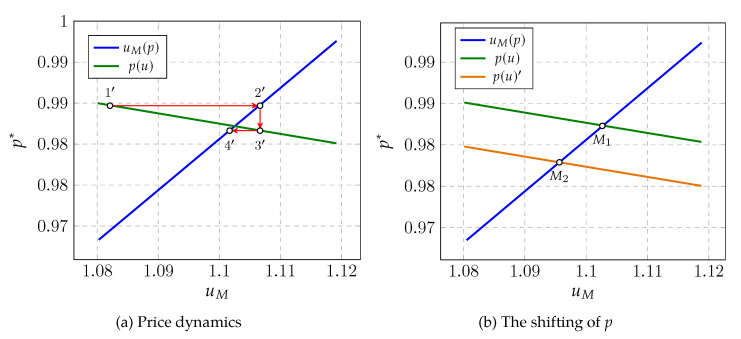
Panic buying.

**Figure 3 ijerph-19-16891-f003:**
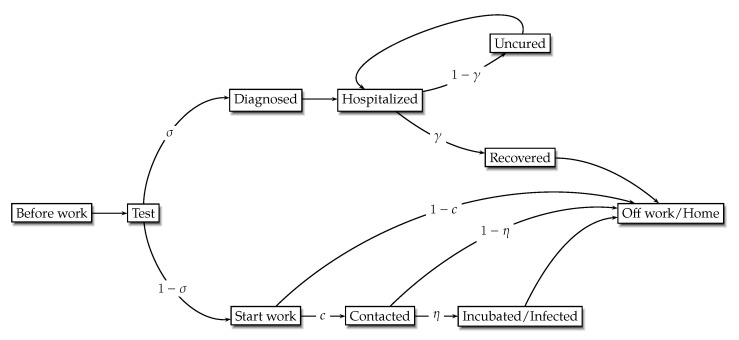
The COVID-19 transmission road map.

## Data Availability

Data available on request due to restrictions.

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
