# Peer review of "A Model of Panic Buying and Workforce under COVID-19"

_ijerph, 2022, doi:10.3390/ijerph192416891_

Round 1

Reviewer 1 Report

Review of An Economic Model of COVID-19

1.       First I don’t believe title is really appropriate. But it is up to authors decide it. Check that title is mentioning COVID-19 economic model and abstract reduce it to retail and buyers relations to medical resources supply. There is something wrong with this rationale in the title or the abstract.

2.       Another point of abstract section is the grammar. Please check it again.

3.       Considering introduction, there is one research that covers government and medical resources during pandemic in Brazil. Please check its suitability to your topic. https://books.google.com.br/books?id=ldZ2EAAAQBAJ&pg=PT39&dq=%22charles+roberto+telles%22&source=gbs_toc_r&cad=3#v=onepage&q=%22charles%20roberto%20telles%22&f=false

4.       ∫h1 di=1, in this equation, there are no explanation about symbols H and d.

5.       Also the first equation detached in the text is not numbered. What means symbols Beta and u?

6.       Concerning masks, which amount of days it can be used by a person? Is it part of the calculus?

7.       One other point, market ability to restock masks is assumed as a prompt continuous production? No possibility of markets be unable to produce due to lack of resources? I mean the shortage can occur not only by the individuals consumption but lack of resources to produce it.

8.       Concerning the mask model at first, the mathematical reasoning would be an ideal mechanism, but concerning item 6 and 7 in this review, would it be a limitation of the study towards masks production?

9.       Also, people compliance to the policies about mask use was evaluated? The study sample of analysis is about China only or it could be considered globally?

10.   Would the item 9 be a limitation of study too?

11.   What about the homemade masks, would it be a downforce factor of the equation about buying and selling equilibrium?

12.   The study model can be applied only to “panic buying” phase or it can be applied continuously to any instance of an epidemic?

13.   In a general sense, this reviewer was going to ask “whats the ideal pandemic or epidemic situation to the model be applied for?”, however, as long as I was reading the article rationale, it seemed to be applied without many of the circumstances I have mentioned before. Also, the article main target of analysis is about workforce availability to produce masks or else. This point is not wrong of course and it is quite difficult to delineate it. Bearing it, I believe article reached a possible conclusion but with several limitations indeed.

14.   One other remark, considering distinct COVID-19 variants and distinct periods of its propagation, and also, as authors state, sub-variant Omicron, which data can sustain authors model relating Omicron to the model? Consider it as the transmission rate, covid test rate, period of infection, incubation, days needed in the hospital, recovery rate,…?

15.   Some points of the study are conclusive and important to worldwide concerns about pandemic. For example, “Second, given the situation that only a fraction of mask (ρt) can be restocked in each period for every household, then for any retail price higher than the reservation value  would arouse the buying competition on masks, resulting in the escalating of mask shortage circumstance.” This was noticed in several countries objectively. However, the main forces affecting this phenomenon are quite varied as some items of this review pointed out. For this, the study needs to add a limitation section where ideal scenario of analysis was considered for the mathematical model, but no other confounding variables were added as well. But its up to authors discretion.

Reviewer 2 Report

1. Some abstract keywords do not appear in the abstract.

2. The authors have not provided any original COVID-19 data, such as demand, supply, and market prices, to support their research article, nor have they provided any of those data that has been tested with a model. It has been demonstrated that their model can be used in a real-world situation. They are only discussing the model's assumptions, and they have proven to only follow the theory.

3: Policy recommendations are flawed. The author must suggest more detail.

Round 2

Reviewer 1 Report

Dear authors, thank you for the modifications where I could see you agree mostly.

Remember these points are important to be mentioned in your manuscript. Also, even the explanations you have made are very important. 

At your discretion, do a final review and check the main points.

Article rationale have improved in terms of possible weak points due to COVID-19 wide variation of factors involved on it among countries.

Thank you.